# Genetics behind Cerebral Disease with Ocular Comorbidity: Finding Parallels between the Brain and Eye Molecular Pathology

**DOI:** 10.3390/ijms23179707

**Published:** 2022-08-26

**Authors:** Kao-Jung Chang, Hsin-Yu Wu, Aliaksandr A. Yarmishyn, Cheng-Yi Li, Yu-Jer Hsiao, Yi-Chun Chi, Tzu-Chen Lo, He-Jhen Dai, Yi-Chiang Yang, Ding-Hao Liu, De-Kuang Hwang, Shih-Jen Chen, Chih-Chien Hsu, Chung-Lan Kao

**Affiliations:** 1School of Medicine, National Yang Ming Chiao Tung University, Taipei 112304, Taiwan; 2Department of Medical Research, Taipei Veterans General Hospital, Taipei 11217, Taiwan; 3Institute of Clinical Medicine, National Yang Ming Chiao Tung University, Taipei 112304, Taiwan; 4Department of Ophthalmology, Kaohsiung Medical University Hospital, Kaohsiung Medical University, Kaohsiung 80708, Taiwan; 5Department of Ophthalmology, Taipei Veterans General Hospital, Taipei 11217, Taiwan; 6Department of Physical Medicine and Rehabilitation, Taipei Veterans General Hospital, Taipei 11217, Taiwan; 7Department of Physical Medicine and Rehabilitation, School of Medicine, National Yang Ming Chiao Tung University, Taipei 112304, Taiwan; 8Center for Intelligent Drug Systems and Smart Bio-Devices (IDS2B), National Yang Ming Chiao Tung University, Hsinchu 300093, Taiwan

**Keywords:** genome-wide association study, phenome-wide association study, genetic diagnosis, pathology, cerebral visual impairment, multiple sclerosis, Joubert syndrome, Mowat–Wilson disease, Zellweger spectrum disorder, neuromyelitis optica spectrum disorder

## Abstract

Cerebral visual impairments (CVIs) is an umbrella term that categorizes miscellaneous visual defects with parallel genetic brain disorders. While the manifestations of CVIs are diverse and ambiguous, molecular diagnostics stand out as a powerful approach for understanding pathomechanisms in CVIs. Nevertheless, the characterization of CVI disease cohorts has been fragmented and lacks integration. By revisiting the genome-wide and phenome-wide association studies (GWAS and PheWAS), we clustered a handful of renowned CVIs into five ontology groups, namely ciliopathies (Joubert syndrome, Bardet–Biedl syndrome, Alstrom syndrome), demyelination diseases (multiple sclerosis, Alexander disease, Pelizaeus–Merzbacher disease), transcriptional deregulation diseases (Mowat–Wilson disease, Pitt–Hopkins disease, Rett syndrome, Cockayne syndrome, X-linked alpha-thalassaemia mental retardation), compromised peroxisome disorders (Zellweger spectrum disorder, Refsum disease), and channelopathies (neuromyelitis optica spectrum disorder), and reviewed several mutation hotspots currently found to be associated with the CVIs. Moreover, we discussed the common manifestations in the brain and the eye, and collated animal study findings to discuss plausible gene editing strategies for future CVI correction.

## 1. Introduction

Cerebral visual impairments (CVIs) represent types of visual disorders characterized by parallel intracranial lesions. Excluding the traumatic and iatrogenic causes of CVIs, 27% of childhood visual impairments in developed countries are described as CVIs [1,2]. These young patients typically present with visual difficulties that cannot be explained by ophthalmological examinations, and in some somatic mutation cases, their condition cannot be traced to their lineage, which generally makes the characterization of CVIs challenging.

To this point, there is a lack of standard approaches to methodology and clear targets in investigating the CVIs. Driven by distinct mutation predispositions, the manifestation of CVIs can vary, and therefore their diagnoses are often made by the serial differential exclusion of mimicking diseases in tandem with a complete set of neuroimaging tests [3]. In this regard, molecular-based diagnostic techniques stand out as an explicit approach, generating clear mutation profiles and helping investigators to narrow down their differential diagnosis [4].

With genome-wide and phenome-wide association studies (GWAS and PheWAS) being gradually adopted to elucidate the genetic predispositions to CVIs, a pattern of mutation ontology has been drawn to classify the CVIs into five categories: (1) ciliopathies (Joubert syndrome, Bardet–Biedl syndrome, Alstrom syndrome); (2) demyelination diseases (multiple sclerosis, Alexander disease, Pelizaeus–Merzbacher disease); (3) transcriptional deregulation (Mowat–Wilson syndrome, Pitt–Hopkins syndrome, Rett syndrome, Cockayne syndrome, X-linked alpha thalassaemia mental retardation); (4) compromised peroxisome disorders (Zellweger spectrum disorder, Refsum disease); and (5) channelopathies (neuromyelitis optica spectrum disorder) (Figure 1).

In this review, we summarize the genetic background and molecular functions at the cellular level that result in brain and eye defects in CVIs. For each distinct disease entity, we discuss the frontier research, such as animal studies, that shed light on either the pathogenesis or therapeutic aspects of the pathology. We summarize the genetic data and discuss the molecular mechanisms underlying the genetically-driven pathology in distinct anatomical locations, such as the CNS in the case of CVIs.

## 2. The Genetic Predisposition to CVIs

Most genetic CVIs are rare diseases with incidence ranging from 1/90,000 to 1/2,700,000 (Table 1). Thus, these types of disorders are less attractive for investment by pharmaceutical companies to study their mechanisms and develop treatment. Therefore, the diagnoses, pathologies, and treatments of these diseases remain largely unknown, which compromises the right of patients to live healthier lives. An association between visual dysfunction and brain diseases was found in many previous studies [5,6]. Fortunately, with the advance of DNA sequencing and information science, there are increasingly more useful approaches to big-data studies in genomics. With a combination of information about genes and diseases, scientists could gain a better understanding of such genetic diseases.

Genetic CVIs are caused by mutations resulting from errors during replication, mitosis, meiosis, or damage without proper repair. These mutations can be classified as missense, frameshifts, or nonsense mutations within coding sequences; other types of DNA alterations may appear as a result of deletions, duplications, or translocations of larger genomic regions. Although the mutation rate is normally about 50–90 de novo mutations per genome per generation in humans [7], few mutations occurring in reproductive cells are passed on to the descendants. Phenotypes with mutations on non-sex chromosomes (autosomes) are not different across genders, while phenotypes with mutations on sex chromosomes (allosomes) lead to sex-linked inheritance. Since genetic CVIs often cluster in families, doctors should be more aware of children and apply treatment earlier.

In order to diagnose genetic CVIs, DNA sequencing is essential. However, the rate of incorrectly identified DNA bases is higher than the frequency of occurrence of genetic mutations [8]. There are three methods to address the error rate problem: barcoding, circle sequencing, and duplex sequencing. In barcoding methods, DNA molecules are amplified by polymerase chain reaction (PCR) after being marked by a barcode, a uniquely identifiable sequence, so the amplified pool can be classified based on this barcode and the mutations then stimulate a higher signal than the PCR errors [9]. In circle sequencing, DNA is denatured into single-stranded forms and circularized; single long reads from rolling circle replication can be computationally split into individual copies of the original circle, and the mutations will then generate a higher signal than replication errors [10]. In duplex sequencing, paired-end reads can be classified into forward (ab-SSCS) or reverse (ba-SSCS) strands after being marked by barcodes on both strands of the DNA that are then reunited into the original duplex consensus sequence (DCS); the mutations can be easily distinguished since the errors will only be present in one strand, which currently makes duplex sequencing the method with the lowest error rate [11]. With improvements in DNA sequencing, the genetic database can be established to make it possible to clarify the gene-disease relationship.

**Table 1 ijms-23-09707-t001:** Epidemiology of Cerebral Visual Impairments.

Type	Disorder ^1^	Subtypes ^2^	Age			Frequency		Male/Female Ratio	Inheritance Mode ^3^
			Onset	Diagnosed	Death (81.8 Years in General Populations [12])	Incidence	Prevalence		
**Ciliopathy**	JBTS	See in Appendix A	10 days~5 months [13]	unknown	7.2 years [14]	1/80,000~1/100,000 [15]	1/80,000~1/100,000 [15,16,17]	1.22 [18]	AR XLR (JBTS10) AD (JBTS19)
	BBS	See in Appendix A	unknown	9 years [19]	25% in 44 years [20]	1/125,000~1/160,000 in Europe population [21,22] 1/65,000 in an Arab population [23]	1/160,000 in European population 1/13,500 in Arabic populations [24]	1.30 [19]	AR AD (BBS1)
	Alstrom Syndrome	-	infancy [25]	unknown	<50 years [26]	1/1,000,000 [27]	1/1,000,000 [26]	0.50 [28]	AR
**Demyelination**	MS	-	18 years~40 years [29]	20 years~50 years [30]	74.7 years [12,30]	2.1/100,000 [31]	35.9/100,000 [31]	0.29~0.91 [32]	autosomal, phantom heritability
	AxD	neonatal	<30 days [33]	unknown	<2 years [33]	1/2,700,000 [31]	1/2,700,000 [34]	0.50 [28]	AD
		infantile	30 days~2 years [33]	unknown	weaks~years [35]				
		juvenile	2 years~12 years [36]	unknown	20 years~30 years [37]				
		adult	>12 years [36]	unknown	decades [37]				
	PMD	-	3 months~9 years [38]	unknown	6 years~25 years [38]	1.45/100,000~1.9/100,000 [39,40]	1/300 000~1/500 000 [41]	>1.00	XLR
**Transcriptional Deregulation**	MWS	-	27.5 months [42]	unknown	<60 years [43]	1/70,000 [44]	1/50,000~1/70,000 [45]	1.00 [46,47]	AD
	PTHS	-	2 years~19 years [48]	unknown	unknown [49]	unknown	1/225,000~1/300,000 [50]	1.00	AD
	RTT^1^	-	4 years [51]	3.5 years [52]	4 years [51]	1/22,800 [53]	1/10,000~1/15, 000 [53]	<1.00 [54]	XLD
	CS	CS type I	0 year~2 years [55]	unknown	16.1 years [55]	1/200,000 [56,57]	2.5/1,000,000 [58]	1.00 [59]	AR
		CS type II	at birth [60]	unknown	5.0 years [60]				
		CS type III	>2 years [60]	unknown	30.3 years [60]				
		XP/CS	0 year~2 years [61]	unknown	7 months~6.4 years [61]				
	ATR-X	-	unknown	unknown	unknown	1/100,000 [62]	1/30,000~1/40,000 [63]	>1.00	XLD
**Compromised Preoxisome**	ZSD	-	0 year~3.8 years [64]	7 days~31 years [65]	depending [64]	1/12,000 in Canadian populations 1/50,000 in US populations 1/500,000 in Japanese populations [66]	unknown	1.00	AR
	RD	ARD	2–7 years [67]	1 year~28 years [68]	4 decades~5 decades [69]	1/250000 [70]	unknown	1.00 [71]	AR
		IRD	early infancy [67]	unknown	5 years~13 years [69]				
**Channelopathy**	NMOSD	-	late fourth decade [72]	unknown	52.3 years [73]	0.053/100,000~0.400/100,00 [74]	1/100,000 in white populations 3.5/100,000 in East Asian populations 10/100,000 in Black populations [75]	0.11~0.43 [76,77]	Multigenic

^1^ JBTS—Joubert syndrome; BBS—Bardet–Biedl syndrome; MS—multiple sclerosis; AxD—Alexander disease; PMD—Pelizaeus–Merzbacher disease; MWS—Mowat–Wilson disease; PTHS—Pitt–Hopkins disease; RTT—Rett syndrome; CS—Cockayne syndrome; ATR-X—X-linked alpha thalassemia mental retardation; ZSD—Zellweger spectrum disorder; RD—Refsum disease; and NMOSD—neuromyelitis optica spectrum disorder. ^2^ XP/CS—xeroderma pigmentosum/Cockayne syndrome; ARD—adult Refsum disease; IRD—infantile Refsum disease. ^3^ AD—autosomal dominant; AR—autosomal recessive; XLD—X-linked dominant; and XLR—X-linked recessive.

## 3. Revisiting CVIs by the GWAS-PheWAS Approach

Genome-wide association studies (GWAS) are defined as observational studies of a genome-wide set of genetic variants in different individuals evaluating variant-disease associations (VDAs), the association between extensive common single nucleotide polymorphisms (SNPs), and disease phenotypes. The first published GWAS successfully found an association of functional SNPs with the susceptibility to myocardial infarction [78]. In addition to finding possible molecular mechanisms of pathology in genetic diseases [79], GWAS may also help in finding differences or similarities between such diseases. For example, neuromyelitis optica spectrum disorders (NMOSD) used to be regarded as a subtype of multiple sclerosis (MS) because of the similar clinical features. However, GWAS studies revealed that the susceptible genetic variants of NMOSD are more similar to systemic lupus erythematosus (SLE) instead of MS [80]. Therefore, GWAS can largely improve the understanding of genetic variants and associated pathological manifestations.

Phenome-wide association studies (PheWAS) are a study design aimed to find the phenotypes that may be associated with a given genetic variant [81]. Since SNPs may influence the expression of more than one gene due to linkage disequilibrium, the issue of gene pleiotropy becomes important, including authentic/horizontal/mosaic/independent pleiotropy and spurious/vertical/relational/reactive pleiotropy. The former indicates multiple independent effects of a mutation that causes multiple phenotypes, while the latter implies multiple effects depending on one another in a cascade that eventually leads to causally related phenotypes [82]. With PheWAS, we may understand the pathologies of rare diseases by investigating other phenotypes caused by the same SNP.

With the thorough understanding of VDAs, doctors may recommend patients with early susceptibility-indicative symptoms to undergo genetic testing, including restriction fragment length polymorphism (RFLP), DNA microarray, whole genome sequencing (WGS), and whole exome sequencing (WES). In such a way, patients may be precisely diagnosed with a particular CVI and receive appropriate treatment before the full manifestation of a disease. RFLP, the first DNA profiling technique inexpensive enough to gain widespread application, detects genetic diseases based on the fragment length of DNA after being cleaved by restriction enzymes and separated by agarose gel electrophoresis. DNA microarray, a conventional method for genetic testing, distinguishes gene mutations by labeling with different fluorescent molecules of the case and control cDNAs [83]. WGS and WES are next-generation sequencing (NGS) techniques with DS technology, which made genetic testing commercial and accessible [84]. For the purpose of reducing the requirement for excessive data analysis and higher cost in WGS and WES, targeted enrichment methods appear to only focus on specific genomic intervals [85]. With the evolution of approaches, the improvement rate of the cost and the quality in genetic testing surpasses Moore’s law [86], the law predicting the growth of technology [87]. Genetic testing thus becomes a practicable method to diagnose rare genetic diseases such as genetic CVIs.

By merging the data from GWAS, PheWAS, and advanced genetic testing approaches, big data can be generated to enable doctors and scientists to re-examine rare genetic diseases on a larger scale (Figure 2). Using this approach, the pathological changes occurring in these rare genetic diseases can be associated with common molecular pathways (Table 2). From a clinical view, proper annotation of molecular pathogenesis in rare diseases may well facilitate accurate diagnosis, comprehensive assessment, on-hit intervention, and prophylactic support remedies. Therefore, possible treatments could be designed for the diseases which were previously ignored due to their low incidence.

## 4. Multiple Sclerosis: A Typical Case of Brain-Eye Parallelism

Multiple sclerosis (MS) is a prevalent genetic disorder that causes brain and eye disability in young populations. MS has a disease onset pattern, in which manifestations in the eye often precede the onset in the brain [176]. In particular, common MS ocular features are optic neuritis (ON) and internuclear ophthalmoplegia [177], while cerebral manifestations are trigeminal neuralgia (TN) and glossopharyngeal neuralgia (GN) (Table 3) [178]. Among these features, the correlation between MS and ON is relatively well-reported: an observational study concluded that half of MS patients have ON [176], and within 10 to 15 years, 34–75% of confirmed diagnosed ON patients develop MS (Table 4) [179,180]. In a similar but independent study on a specialized ON cohort (*n* = 115) at Moorfields Eye Hospital, it was shown that patients with new MRI T2 brain lesions within 1 year of ON diagnosis were likely to develop concurrent MS in the following three years (prediction: 85% sensitivity and 79% specificity) [181]. Taken together, the coupled manifestations between the brain and the eye are evident, and they conform to certain chronological patterns in clinical presentation. 

To elucidate the molecular mechanisms that contribute to the association between ON and MS [136,182,183], investigation approaches such as histology biomarker measurements, establishing MS animal models, and cell-based transcriptome analysis were conducted. Nevertheless, the major MS–ON correlation finding was discovered by the conjunction of GWAS sequencing and PheWAS characterization. Particularly, the SNP variants of MS–ON patients were highly enriched in the genomic regions encoding human leukocyte antigen (HLA) family members [135], major histocompatibility complex (MHC) [136], inflammasome units [137], and members of complement pathways [79]. These SNPs may affect the immune system-related pathways and thus underly the demyelinating features of the central nervous system (CNS) and the neuritis in the retina. Such studies follow a common scenario in which the genetic finding gives clues to molecular mechanisms in clinical pathology. In MS, the genetic profile of inflammation genes could predict ocular manifestations; for instance, the C3 mutations were associated with ganglion cell/inner plexiform layer atrophy (*p* = 0.004) in the retina; meanwhile, C1QA and CR1 gene mutations were associated with low-contrast letter acuity (LCLA) loss, a hallmark ocular manifestation in MS. In addition to the eye, C3 gain-of-function mutation rs2230199 has also been linked to lower brain volumes [184,185,186], indicating that the same set of inflammation gene mutations have prediction values in both the brain and the eye manifestations of the MS patients.

To link these genetic findings to the actual molecular pathogenesis mechanisms, animal models were established to validate the causal effects of gene SNP candidates. Experimental autoimmune encephalomyelitis (EAE) is a frequently used MS-mimicking mouse model, in which C3 pathway upregulation was found in the neurotoxic astrocytes of the brain shrinkage region, a consequence of demyelinating neuron atrophy [187]. To confirm the role of the C3 pathway as a direct contributor to MS development, several studies were performed to knock out C3 genes in the EAE mouse model. Importantly, the ablation of the C3 pathway reduced T cell infiltration and inflammatory cytokine production in these MS-conditioned mice [188]. It is worth mentioning that C3 protein expression was found to be more elevated in the female mice astrocytes than in the litter-mate male control, this pattern fits the clinical finding of female MS patients subjects exhibiting worse neurology symptoms. Additionally, female EAE mice were found to be subject to more severe retinal ganglion cell (RGC) axon loss than the male EAE control, revealing a negative correlation between C3 expression level and the RGC axon length (r = −0.64, *p* = 0.04) [187]. Using this approach, EAE animal models validated the role of C3 in MS, and meanwhile highlighted both the pathological (brain atrophy and retina axon loss) and epidemiological (gender subjectivity) clinical features of MS.

To summarize, the investigation of correlations between C3 and MS is an example of a combined genetic and animal study effort to characterize the disease pathogenesis in a collaborative but independent manner. Nevertheless, in addition to MS, there are plenty of other brain-eye CVI diseases which also show parallelism [189,190,191,192,193], but have yet to be fully studied (listed as in Table 4). In the following section of this review, we summarize other CVI diseases by addressing such details as documented phenotypes, genetic diagnosis, and histology findings with molecular characterization.

**Table 3 ijms-23-09707-t003:** Brain-Eye Correlations in Cerebral Visual Impairments.

Type	Disorder ^1^	Cerebral ^2^	Visual ^3^
**Ciliopathy**	JB-Ret	MTS (100%) [194,195] developmental delay (100%) mental retardation (100%) hypotonia (100%) [196] Dandy-Walker malformation (10%) [197]	RP (100%) [198] ocular motor apraxia (80%) strabismus (74%) nystagmus (72%) [199] RD (30%) [198] chorioretinal coloboma (30%) optic nerve atrophy (22%) [199]
	BBS	developmental delay (50–91%) ataxia (40–86%) [200] cognitive impairment (66%) central obesity (89%) [201] functional independence (74%) attention capacity (69%) [202] ID (62%) [203] perceptual reasoning (53%) verbal fluency (22–44%) [202]	RD (94%) RP (43%) [201]
	Alstrom Syndrome	developmental delay (45%) [204]	RD (100%) [27] blind (90%) [205]
**Demyelination**	MS	Dawson’s fingers (92.5%) [206] central pain (15–85%) [207] central trigeminal involvement (12–38%) [208] braunstem dysfunction (25%) sensory disturbances (18.3%) motor disturbances (17.5%) [209] TN (6%) [210] ID (2%) [211]	ON (50%) [176] abnormal blink reflex (89%) [212] nystagmus (10%) [213]
	AxD	bulbar sign (83.3%) changes in lower brain stem or upper cervical cord (82.4%) [214] cerebral white matter lesions (80%) [215] pyramidal sign (63.4%) changes in cerebellum or dentate hilum (54.1%) ataxia (50%) dysarthria (42%) [214] mental retardation (29.0%) epilepsy seizure (26.5%) pseudobulbar sign (21.6%) [216] cyst formation (25%) [217] changes in basal ganglia or thalami signal (17.6%) autonomic disturbances (11.4%) macrocephaly (9.8%) cranial sensory disturbances (6.8%) [216]	ocular motor abnormalities (46.1%) [216] nystagmus (33%) [214]
	PMD	developmental delay (100%) corpus callosum atrophy (100%) hypotonia (83.8%) displayed supratentorial brain atrophy (29.0%) pyramidal sign (5.4–22.2%) epilepsy seizure (7.1–14.3%) ataxia (5.4–7.4%) cerebellum atrophy (3.2%) [193]	nystagmus (99.1%) [193]
**Transcriptional Deregulation**	MWS	hypotonia (93%) [47] microcephaly (81%) [141,218,219,220,221,222] neocortical projections (79.6%) hippocampal abnormalities (77.8%) enlargement of cerebral ventricle (68.5%) [47] epilepsy seizure (73%) [141,218,219,220,221,222] brain anomalies (43%) reduction of white matter thickness (40.7%) localized signal alterations of the white matter (22.2%) [222]	eye anomalies (4.1%) [141,223]
	PTHS	ID (98%) gross motor development (92%) hypotonia (69%) ataxia (57%) [141] epilepsy seizure (40%) small corpus callosum (23%) enlargement of cerebral ventricle (21%) microcephaly (17%) [50]	strabismus (45%) myopia (39%) [49] astigmatism (26%) [50] nystagmus (4%) [49]
	RTT	deceleration of head growth (80%) epilepsy seizure (60~80%) [224] language disorder (61.5%) microcephaly (46.2%) gross motor development (30.8%) [225]	difficulty recognizing unfamiliar things [226] selectively focused on specific things [227] vision search difficulty [228]
	CS	abnormal myelination in brain (93%) [61] mental retardation (90%) microcephaly (83%) motor disturbance (71%) [229] tremor (66%) [190] intracranial calcifications (63%) [61] ventricular dilatation (23%) [191] epilepsy seizure (5–10%) [230]	RP (60–100%) [231] RD (33–89.3%) [58] cataracts (15–36%) [231]
	ATR-X	developmental delay (100%) ID (100%) [232] language disorder (95%) [233] hypotonia (80–90%) [232] microcephaly (75%) [233] brain atrophy (63%) high intensity of white matter (41%) [232] epilepsy seizure (30–40%) [232] delayed myelination (15%) [63]	ocular defects (25%) [234]
**Compromised Preoxisome**	ZSD	peripheral neuropathy (58%) T2 hyperintensities (50%) cerebellar sign (47%) cerebellar cortical atrophy (38%) pyramidal sign (26%) high intensity of white matter (25%) hypotonia (21%) [65]	VA disability (100%) RP (84%) retinopathy (84%) night blindness (84%) retinal degeneration (63%) [65]
	RD	polyneuropathy (70%) ataxia (50%) [192]	RP (100%) [192] pupils (78%) VA (76.7 %) visual fields (75%) cataracts (30%) nystagmus (22%) glaucoma (17%) [68]
**Channelopathy**	NMOSD	periependymal lesions (75%) [235] central vomiting (65.38%) central hiccups (50.00%) pyramidal tract sign (42.31%) [236] LETM (32.9%) brainstem symptoms (4.5%) [237]	ON (22.4%) [238] ophthalmoplegia (19.23%) MLF syndrome (11.54%) [239]

^1^ JBTS—Joubert syndrome; BBS—Bardet–Biedl syndrome; MS—multiple sclerosis; AxD—Alexander disease; PMD—Pelizaeus–Merzbacher disease; MWS—Mowat–Wilson disease; PTHS—Pitt–Hopkins disease; RTT—Rett syndrome; CS—Cockayne’s syndrome; ATR-X—X-linked alpha-thalassaemia mental retardation; ZSD—Zellweger spectrum disorder; RD—Refsum disease; and NMOSD—neuromyelitis optica spectrum disorder. ^2^ MTS—molar tooth sign; TN—trigeminal neuralgia; ID—intellectual disability; and LETM—longitudinally extensive transverse myelitis. ^3^ RD—retinal dystrophy; RP—retinitis pigmentosa; ON—optic neuritis; VA—visual acuity; and MLF—medial longitudinal fasciculus.

**Table 4 ijms-23-09707-t004:** Brain-Eye Parallelism in Cerebral Visual Impairments.

Disease ^1^	Brain MRI Description	Cerebral Disorders ^2^					Visual Disorders ^2^					Disease Process ^3^
		ID	Epilepsy Seizure	Hypotonia	Ataxia	Microcephaly	Nystagmus	RP	RD	Cataracts	ON	
**Ciliopathy**												
JBTS	Molar Tooth Sign (MTS) on T2 MRI			·			·	·	·			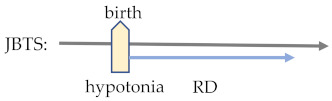
BBS	Shrinkage of the hippocampus and striatum	·			·			·	·			
Alstrom Syndrome	Increased white matter density and small leaks near the ventricles on T1 MRI								·			
**Demyelinating**												
MS	Finger-shaped lesion in the corpus callosum on T2 MRI	·					·				·	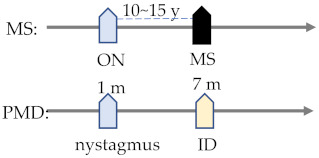
AxD	Shrinkage of the medulla oblongata and the upper spinal cord	·	·		·		·					
PMD	Corpus callosum shrinkage	·	·	·	·		·					
**Transcriptional Deregulation**												
MWS	Corpus callosum hypoplasia, abnormal hippocampus, ventricular enlargement		·	·		·						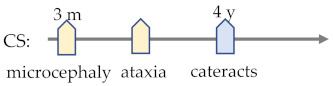
PTHS	Corpus callosum hypoplasia, ventricular enlargement	·	·	·	·	·	·					
RTT	Shrinkage of the corpus callosum and the cerebellum, brainstem narrowing	·	·			·						
CS	Calcification		·			·		·	·	·		
ATR-X	Brain shrinkage, ventricular enlargement	·		·		·						
**Compromised Peroxisome**												
ZSD	T2 hyperintensity	·		·				·				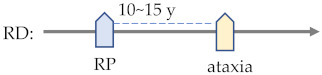
RD	Increased white matter density near the ventricles on T2 MRI				·		·	·		·		
**Channelopathy**												
NMOSD	Marbled lesions above the corpus callosum	·									·	

^1^ JBTS—Joubert syndrome; BBS—Bardet–Biedl syndrome; MS—multiple sclerosis; AxD—Alexander disease; PMD—Pelizaeus–Merzbacher disease; MWS—Mowat–Wilson disease; PTHS—Pitt–Hopkins disease; RTT—Rett syndrome; CS—Cockayne syndrome; ATR-X—X-linked alpha-thalassaemia mental retardation; ZSD—Zellweger spectrum disorder; RD—Refsum disease; and NMOSD—neuromyelitis optica spectrum disorder. ^2^ ID—intellectual disability; RD—retinal dystrophy; RP—retinitis pigmentosa; and ON—optic neuritis. ^3^ There are some common cerebral and visual disorders (expressed in yellow and blue respectively) shared by different cerebral visual impairments (CVIs) that occur at different timings in the disease’s process [189,190,191,192,193]. By revealing molecular mechanisms underlying these clinical features, physicians will be able to diagnose pathologies and apply treatment before the appearance of symptoms.

## 5. Ciliopathy

Ciliopathy is a disease category that affects cells and organs with high demands in cytoskeleton turnover. Neuron cells in particular require robust vesicle transportation, axon extension, and dendrite formation/connection. Ciliary gene mutations result in defects of the transition zone (TZ) complex, an anchor structure that lies between the axoneme and the basal body, and play a crucial role in ciliogenesis and ciliary membrane composition [239,240]. For instance, the Abelson helper integration site 1 (AHI1) gene encodes a protein located in the BB of the primary cilia that interacts with Huntingtin-associated protein 1 (HAP1) to facilitate cerebellar and brainstem development [241]. Concurrently, AHI1 mutations have been reported to cause retinal dystrophy [242], a consequence of the abrogated secretion of neurotrophic factors and impaired rhodopsin trafficking in the eye. Since AHI1 function is implicated in both the development of the brain and the homeostasis of the eye, its mutations result in clinical features in similar, but not identical, CVI diseases such as Joubert syndrome and Bardet–Biedl Syndrome. 

The importance of ciliary genes was also highlighted in their role in intra- and extra-cellular communication. Specifically, the ciliary microtubules interact with components of several signaling pathways. In such a way, the cytoskeleton systems are synchronized with such signaling pathways as the Sonic Hedgehog (SHH), Wingless and Int-1 (WNT), and the G protein-coupled receptor (GPCR). Moreover, these pathways are essential players in the regulation of neural development. The SHH pathway is essential for triggering midbrain dopaminergic (mDA) neurons generated in the ventral midbrain [243]. WNT pathway aberrations have been linked with cerebellar vermis fusion/hypoplasia [244]. GPCR pathways are mainly studied in a photoreceptor cellular context [245,246]. For instance, the first 2.8 Å resolution crystal structure of the GPCR family protein was established by the model of rhodopsin binding to its substrate, 11-cis retinal [247]. On the whole, the ciliary gene defects may lead to both significant cerebral and retinal disorders. 

### 5.1. Joubert Syndrome

Joubert syndrome (JBTS) and Joubert syndrome-related disorders (JSRD) are the conditions presenting as agenesis of the cerebellar vermis, which is identified by the molar tooth sign (MTS) on the MRI (Table 4). 

Since the first gene responsible for JBTS, NPHP1, was identified in 2004 [107], over 40 genes have been found to be associated with this condition, as summarized in the OMIM database (Appendix A). Nonetheless, most of these genes each account for less than 10% of cases, as the genes identified thus far are implicated in the functions of the subcellular structure and the primary cilium. On the molecular level, the genes play a role at the transition zone (TZ) (JBTS2, JBTS4-7, JBTS9, JBTS14, JBTS16, JBTS18, JBTS20, JBTS21, JBTS24, JBTS28, JBTS29, and JBTS34), in SHH signaling (JBTS8, JBTS10, JBTS12, JBTS17, JBTS18, JBTS21, and JBTS23), basal body (BB) (JBTS3, JBTS15, JBTS26, JBTS30, JBTS31, and JBTS33), and in other functions of the primary cilia (JBTS1, JBTS13, JBTS19, JBTS22, JBTS25, JBTS27, JBTS32, and JBTS35-40). The primary cilium is an essential regulator of numerous signaling pathways essential for the movement of cells and fluids in response to sensory inputs involved in photoreception [248], so patients with JBTS exhibit co-manifestation of brain and eye symptoms. 

Additionally, robust evidence has linked ciliary machinery to GPCR functions that affect brain and retina health. From the preliminary findings of neural development studies in mice, deleting JBTS-related genes Arl13b and Inpp5e implicated in cilia sensing apparatus may abrogate GPCR signaling. Without a proper GPCR signaling cascade, the abrogated PI3K/AKT transcription response leads to the defasciculation and misorientation of brain axon projections [249]. Likewise, in a mouse model of retina-specific Inpp5e knockout (Inpp5e^F/F^; Six3-Cre), Inpp5e loss impairs photoreceptor axoneme formation and emulates the optic disc dysmorphism seen in JBTS patients [250]. Besides Inpp5e and Arl13b mutation, malfunctioned cilia genes such as CELSR2 [251] and KIAA0586 (TALPID3 in chicken) [252] were also discovered to contribute to JBTS-related symptoms. On the whole, a considerable number of gene mutation studies affecting the primary cilia components have given clues for both in vitro and in vivo pathogenic mechanisms of the retina and brain defects commonly seen in JBTS patients. 

Clinically, JBTS is often associated with other diseases, such as oral-facial-digital (OFD) syndrome, acrocallosal (AC) syndrome, Jeune asphyxiating thoracic dystrophy (JATD) features, and retinal, renal or hepatic diseases [253]. In terms of manifestation of CVI-related features, JBTS can be classified into pure JBTS (JBTS8, JBTS13, JBTS15, JBTS22, JBTS25-27, JBTS30, JBTS32, JBTS33, and JBTS35-40), mixed JBTS with retinal diseases (JBTS1-5, JBTS7, JBTS9, JBTS14, JBTS16, JBTS20, JBTS28, and JBTS29), and mixed JBTS without retinal diseases (JBTS6, JBTS10, JBTS12, JBTS17-19, JBTS21, JBTS23, JBTS24, JBTS31, and JBTS34). Pure JBTS has 3 main diagnostic features: MTS, hypotonia in infancy with later ataxia, and developmental delays [131]. JBTS with retinal disease (JS-Ret) is characterized by pigmentary retinopathy similar to classic retinitis pigmentosa, and 30% of JS-Ret patients develop retinal dystrophy [198]. Additionally, clinical features of JB-Ret also include optic nerve atrophy (22%), chorioretinal coloboma (30%), oculomotor-related symptoms (ocular motor apraxia (80%), strabismus (74%), and nystagmus (72%)) (Table 3) [199]. Since most of these symptoms are progressive with age and evolve over time, regular monitoring is essential to ensure diagnosis and treatments on time.

### 5.2. Bardet–Biedl Syndrome

Bardet–Biedl syndrome (BBS) is a ciliopathy affecting multiple systems, including cognitive impairment (66%), central obesity (89%), and retinal dystrophy (94%) [201] (Table 3). A group of proteins named after this syndrome (BBS proteins) participate in the functions of primary (non-motile) cilia via the BBSome complex (formed by BBS1, BBS2, BBS4, BBS5, BBS7, BBS8, BBS9, and BBS18), chaperonin complex (formed by BBS6 and BBS12), and IFTB (formed by BBS19, BBS20, and BBS22) (Appendix A). The BBSome complex regulates cargo delivery in primary cilia at TZ via two signaling pathways: (1) vesicular sorting from the Golgi complex via interacting with the Rabin8 (Rab8 GDP/GTP exchange factor); (2) selective transportation along the cilium via acting as an adaptor between cargo and intraflagellar transport (IFT) particles [254]. The chaperonin complex mediates BBSome assembly at centrosomes [255] and basal bodies [256] by stabilizing the first component BBS7 under the regulation from BBS10 [257], and by acting as an intermediate for the binding between BBS7 and chaperonin containing TCP-1 (CCT) [258], which accomplishes the following assembly. IFTB mediates anterograde trafficking powered by the kinesin-2 motor. Other BBS proteins, such as BBS14, BBS15, and BBS16, function on basal bodies by recruiting the BBSome complex. Because primary cilia regulate the development of the CNS by sensing local environmental signals and promoting neuronal proliferation and differentiation, dysfunctional BBS proteins impair brain and retina health by deregulating primary cilia.

Defects in BBS proteins influence hippocampal development and neurogenesis signaling [259], which impairs consolidation from short-term memory to long-term memory. Patients with BBS exhibit impairments in intellectual functions (20–25%), verbal fluency (22–44%), perceptual reasoning (53%), attention capacity (69%), and functional independence (74%) [202]. Moreover, leptin receptors located on the ciliary membrane in hypothalamic neurons affect hunger and energy use by regulating adipose tissue mass, so transient ciliogenesis causing adipogenesis is one of the reasons for obesity in BBS patients [256]. In the retina, the connecting cilium connects inner and outer segments of photoreceptors, along which proteins of the phototransduction cascade (arrestin and the visual G protein transducin) move in response to light stimulation. Abnormal trafficking across the defective cilia causes retinal dystrophy [254], which leads to retinitis pigmentosa (RP) or eventually causes night blindness or even complete blindness [260]. 

Since BBS is a multisystem syndrome affecting the brain, retina, and endocrine or metabolic systems, and causes obesity, treatments mainly focus on symptom management. Recently, the FDA approved the first drug, setmelanotide (IMCIVREE), to improve weight management in BBS [261]. Although gene therapy has achieved success in treating other ocular genetic diseases such as Leber congenital amaurosis (LCA) with FDA-approved treatment [262], a pre-clinical gene therapy study in mouse models based on BBS1 overexpression in the wild type retina was shown to cause cytotoxicity [263]. Therefore, gene therapy may not be suitable for BBS.

### 5.3. Alstrom Syndrome

Alstrom syndrome is a rare autosomal recessive genetic disorder caused by nonsense and frameshift mutations primarily in exons 8 (21%), 10 (23%), and 16 (40%) of the ALMS1 gene (Table 2) [134]. Although the precise functions of ALMS1 remain unknown, it is believed that it is involved in ciliogenesis and the centriolar stability of primary cilium function [264]. Cells with ALMS1 knockdown have longer primary cilia with abnormal morphology, including axonemal segmentation, ciliary bulging, and ciliary bending [265]. The depletion of ALMS1 in cells diminishes transforming growth factor beta/bone morphogenetic protein (TGF-β/BMP) signaling [264], a central regulator of cell proliferation, differentiation, and survival programs regulated by the primary cilium [266]. The depletion of ALMS1 in the cells also diminishes the CNAP1 level [264], the centrosome cohesion protein essential for the linkage between the two basal bodies formed by rootletin fibers [267]. Therefore, mutations of ALMS1 may affect primary cilia, threatening brain and retina health.

Since ALMS1 is found in centrosomes, basal bodies, and cytosol in many organs, including retinal photoreceptors and the brain, the progressive development of multi-organ pathologies of Alstrom syndrome includes retinal dystrophy, neurosensory deficit, and type 2 diabetes mellitus. Abnormal primary cilia caused by ALMS1 mutation lead to defects in the transportation of vesicles or disruption of the exocytosis mechanism, causing accumulation of rhodopsin vesicles [268]. Furthermore, primary cilia with the knockdown of ALMS1 cause defects in the multipolar–bipolar transition and retarded neuronal migration [269]. The reduction of neurons caused by the absence of ALMS1 in basal bodies of hypothalamic neurons in mutated mice causes deregulation of appetite [270]. Therefore, the treatment of Alstrom syndrome is complex and individualized due to the combination of multiple disorders. 

## 6. Demyelinating Diseases

Demyelinating diseases can be categorized by their etiology into extrinsic (caused by toxic, chemical, or autoimmune cues) demyelinating myelinoclastic diseases, and intrinsic genetic-mutated demyelinating leukodystrophy. By such means, MS is clastic, whereas other demyelinating diseases, such as Alexander disease (AxD) and Pelizaeus–Merzbacher disease (PMD), which affect the brain and eye, are leukodystrophic.

### 6.1. Alexander Disease

90% of currently-identified Alexander disease (AxD) cases are characterized by cellular gain-of-function glial fibroblast acidic protein (GFAP) aggregates, also known as Rosenthal fibers, in the astrocytes [138,139]. The symptoms of the disease include seizures, spasticity, delayed brain and nystagmus development, impaired oculopalatal myoclonus, and saccadic dysmetria of the eye (Table 3) [271]. Recently, as was found and established in the post-mortem brain and induced pluripotent stem cell (iPSC)-derived astrocytes of AxD patients, GFAP accumulation resulted in the upregulated secretion of chitinase-3-like protein 1 (CHI3L1). Although CHI3L1 does not possess chitinase activity to hydrolyze chitin substrates, the protein is widely found to be associated with tissue inflammation, extracellular matrix (ECM) remodeling, fibrosis, and bronchial asthma. Moreover, clinical evidence has tightly linked CHI3L1 to the features of neurodegenerative diseases. Elevated CHI3L1 levels can be detected in the cerebral spinal fluid (CSF) of Alzheimer’s disease patients [272], post-mortem spinal cord biopsy [273] of amyotrophic lateral sclerosis [274], serum and CSF of MS patients [275,276,277], and the RNA-Seq data from the dorsolateral prefrontal cortex biopsies of schizophrenia patients [278]. In the iPSC-derived neuron cell co-culture experiments, CHI3L1 produced from the GFAP-laden astrocytes was recognized as a neuro-paracrine mediator that inhibits the proliferation and myelination of the oligodendrocyte progenitor cells [279]. The demyelinating effects and GFAP aggregates in astrocytic cytoplasm cause variable neurological symptoms, resulting in AxD. In addition to the brain involvement in AxD, recent studies have shown the association of CHI3L1 with aberrant autophagy in the retinal pigment epithelium (RPE) by activating AKT/mTOR and ERK pathways in rat models, indicating the possible consequence of eye symptoms [280].

The disease has been categorized into infantile, juvenile, and adult forms according to the respective timing of disease onset, and neurodegenerative symptoms can be observed from the prenatal period through to adulthood. However, especially in late-onset AxD, patients’ conditions are frequently complicated by oculomotor defects. This may be confusing as GFAP dysregulation in the ocular context is often linked with stress response in the Muller cells, which may perturb the photosensory retina but not the oculomotor function. Although direct evidence has yet to be obtained on clinical or post-mortem AxD patient samples, the GFAP-positive astrocytes have been known to form the glia limitans at the junctions between the motor neuron rootlet outlet and the CNS nuclei [281]. GFAP-positive astrocytes, as seen by confocal microscopy, form seals at the interlining of the oculomotor nerve and the cortex boundary [282]. In early animal studies using the Gallotia galloti lizard to elucidate the roles of GFAP in midbrain development, Monzon-Mayor, et al. found that GFAP-positive radial glia, together with GFAP-positive astrocytes, preferentially locate in the periphery of the marginal optic tract and the oculomotor nuclei [283]. Therefore, it was speculated that the GFAP-positive glia and astrocytes may be involved in oculomotor nuclei wiring with oculomotor nerves in midbrain development. 

### 6.2. Pelizaeus–Merzbacher Disease

Pelizaeus–Merzbacher Disease (PMD) is a demyelinating disease characterized by oligodendrocyte impairment in the CNS system. The broadly-affected brain areas lead to both psychomotor retardation and spastic quadriparesis in the brain as well as nystagmus in the eye. PMD is caused by mutations in a myelin component gene proteolipid protein 1 (PLP1). PLP1 normally encodes a full-length PLP1 protein and a truncated product DM20 which are both important for myelin formation (Table 2). When mutated, distinct types of PLP1 and DM20 malfunctioned proteins contribute to varying degrees of PMD-associated demyelination; the degree of brainstem demyelination on T2 MRI can distinguish the severe PMD form from the mild forms (Table 4); other detailed radiology correlations with PMD have been reviewed elsewhere [284]. From the genetic perspective, the complete loss (due to deletion and early premature stop codon) of PLP1 and DM20 often gives rise to mild PMD phenotypes [140], whereas the missense mutation types of PLP1 and DM20 elicit a robust unfolded protein response UPR [285] and result in more severe phenotypes of PMD [286]. As shown by in vitro and in vivo experiments, the defect of proper folding as well as the overproduction of PLP1 may lead to excess endoreticular stress and cell death in oligodendrocytes [287,288]. Moreover, PLP1 dysfunction may lead to the accelerated turnover of myelin structure, henceforth enhancing the recycling of myelin lipids. Lipid peroxidation and its subsequent stress-induced cellular sensitization toward free iron implicates the role of ferroptosis in the death of PMD oligodendrocytes [289]. On the whole, the elucidation of the role of GFAP aggregates and PLP1 accumulation in AxD and PMD pathogenesis, respectively, has facilitated research into therapeutic strategies.

### 6.3. Possible Treatments

Given the deleterious consequences of GFAP overexpression in AxD, a research group from the Waisman Center at the University of Wisconsin-Madison applied intracerebroventricular (ICV) bolus injection of an antisense oligonucleotide (ASO) targeting Gfap transcript in Gfap^+/R236H^ mice. Merely 8 weeks post-injection, GFAP protein content in the mutants dropped to an undetectable level in the region of the hippocampus and olfactory bulb. This promising observation is indicative of the rapid turnover of GFAP by the proteasome system that lasts for a period sufficient to show improved body condition in mutant mice [290]. Similarly to the ASO approach that restricts GFAP expression at a transcriptional level, RNAi-mediated inhibition of Gfap may potentially ameliorate retinal and CNS degeneration [291]. Another strategy to prevent GFAP aggregation reported by Bargagna-Mohan et al. showed that withaferin A could covalently bind GFAP and downregulate its expression in Muller cells in the mouse retina [292]. The cellular stress caused by abnormal PLP1 in PMD was addressed by a number of studies to design stress-countering strategies to minimize the detrimental effects caused by defective PLP1 and myelin metabolites [46,289,293]. Notably, a recent study using PLP1-targeting ASO has demonstrated a prominent therapeutic effect in the PMD patient iPSC-derived oligodendrocytes as well as the PMD-mimic Plp1 mutant Jimpy mice [293]. The PLP1-targeting ASO rescued a number of MYRF-stained oligodendrocytes in the brain regions of corpus callosum, cerebellum, and brainstem. Plp1 mutant Jimpy mice receiving one dose of PLP1 ASO exhibited reduced seizure onset and gained resistance to hypoxia condition, indicating that PLP1 ASO treatment could improve the neural commands on respiratory functioning in the PMD mice. In parallel, another research group successfully restored proper myelin formation in oligodendrocytes by applying systemic iron chelator deferiprone to the iron-intoxicated Plp1 mutant Jimpy mice [289]. 

## 7. Transcriptional Deregulation

### 7.1. Mowat–Wilson Syndrome

Mowat–Wilson syndrome (MWS) is a multiple congenital anomaly syndrome caused by mutations in the zinc finger E-box-binding homeobox 2 gene (ZEB2, also called ZFHX1B) (Table 2). The initiation codon is located in exon 2, the stop codon is in exon 10, and mutations most frequently occur in exon 8 (58%) [141], which comprises around 60% of the coding sequence [294]. The heterozygous pathogenic variant is more common (84%) and results in milder forms of MWS, while the heterozygous deletion is less common (15%) but results in an earlier onset [43]. ZEB2 is a DNA-binding zinc-finger transcription repressor targeting 5′-CACCT sequences in different promoters, interacting with activated SMADs, transducers of TGF-β signaling [295], and NuRD complex [296], so it is also called the SMAD-interacting protein-1 (SMADIP1 or SIP1). TGF-β is a superfamily of signaling molecules acting on membrane receptors to influence embryogenesis and control neural development, including bone morphogenetic proteins (BMPs). SMAD proteins are a family of intracellular effectors downstream of TGF-β that trigger the phosphorylation of cytoplasmic molecules. NuRD is a nucleosome-remodeling and histone deacetylation complex repressing E-cadherin (CDH1), the gene which is involved in developmental processes and also targeted by ZEB2 [297]. Therefore, mutations of ZEB2 directly or indirectly influence gene expression.

Clinical features of MWS include epilepsy seizures (79%), microcephaly (78%), CNS anomalies (68.5%), strabismus (50%), and structural eye anomalies (10%) (Table 3) [42]. Epilepsy seizures result from a complex network of pathological events caused by the hypersynchronized activity of neurons and dysfunctional inhibitory interneurons [298]. Both cortical and striatal interneurons are generated from medial ganglionic eminence (MGE) [299]. A homeobox domain-containing transcription factor NKX2-1, which is normally repressed by ZEB2 after immature interneurons migrate from MGE to the cortex, causes interneurons to differentiate into normal cortical neurons. In the absence of ZEB2, NKX2-1 induces transformation of interneurons into striatal nNOS/NPY/Sst GABAergic interneurons [300]. The diverse population of GABAergic interneurons increases the risk of epilepsy by disturbing the balance between excitation and inhibition [298].

CNS anomalies often present as hippocampal abnormalities (77.8%), neocortical projections (79.6%), white matter (40.7%), and enlargement of cerebral ventricles (68.5%) [221]. Hippocampal interneurons are also MGE-derived, indirectly regulated by ZEB2. The microtubule minus-end binding protein ninein is regulated by ZEB2 and stabilizes microtubules and accelerates their growth, thereby enhancing the formation of neocortical axons [301]. Neocortical axons include intercortical connections (corpus callosum (CC)), anterior commissure (AC)), corticofugal projections (corticothalamic tract (CT)), and corticospinal tract (CST) [302,303]. Moreover, since ZEB2 promotes the transition from immature to mature myelinating oligodendrocytes by antagonizing BMP receptor-activated SMAD activity [304], mutations of ZEB2 lead to the reduction of white matter thickness (40.7%) and localized signal alterations of the white matter (22.2%) [221]. Furthermore, neurotrophin-3 (NTF3) and fibroblast growth factor 9 (FGF9), the signaling factors normally repressed by ZEB2, can feed back from postmitotic neurons to progenitors, to regulate the timing of maturity and the number of neurons and glial cells throughout corticogenesis [305], thereby influencing the volume of cerebral ventricles. ZEB2 is also expressed in the neural retina [306] and whole lens [307]. The interactions between ZEB2 and TGF-β influence the development of neural crest-derived cells [308], causing structural eye anomalies including microphthalmia, retinal colobomas, axenfeld anomaly, ptosis, cataract, and retinal aplasia [42]. Without advanced genetic testing, MWS case numbers are likely under-reported, since the survival of MWS patients into adulthood up to as much as 60 years old is possible [43]. Nonetheless, there is no specific treatment for MWS because the defects result from mutations affecting embryonic development [307].

### 7.2. Pitt–Hopkins Syndrome

Pitt–Hopkins syndrome (PTHS) is known by the TCF4 mutation and chromosome 18 aberrations (Table 2). Common presentations of PTHS patients are severe motor and mental retardation, typical facial features, and breathing anomalies, which share phenotypic similarities to Angelman syndrome (associated with UBE3A) and Mowat–Wilson syndrome (MWS) (associated with ZEB2) (Table 3) [309]. TCF4 is a histone deacetylase that regulates the chromatin structure and transcription during neurogenesis. Transcriptome studies of TCF4 knockdown in neuroblastoma cell lines have demonstrated the abrogation of several pathways, such as in EMT and TGF-β signaling. The link of TCF4 defects with neurogenesis has been explored in neural precursor proliferation and differentiation as well as axonal migration and dendritogenesis in synapse formation. In GFAP-cre::Tcf4Fl/Fl neural-specific Tcf4 knockout mice, neural precursor cells experienced a severe differentiation delay and displayed shortened apical dendrites with increased branching [142]. TCF4 was found to bind MATH1, a proneural transcription factor of rhombic lip progenitor neurons that were required for hindbrain establishment [310]. Additionally, TCF4 is highly expressed in the adult hippocampal dentate gyrus, one of the few brain regions where neural stem/progenitor cells generate new functional neurons throughout life. Scientists also assayed whether histone deacetylase (HDAC) inhibition would be sufficient to normalize the enhanced hippocampal long-term potentiation (LTP) phenotype [311]. Surprisingly, treatment with the HDAC inhibitor trichostatin resulted in significantly reduced LTP in the Tcf4^+/−^ mouse hippocampus. Moreover, Hdac2 knockdown and subchronic treatment with HDAC inhibitor suberoylanilide hydroxamic acid (SAHA) were sufficient to improve learning and memory in Tcf4^+/−^ mice, thus indicating that cognition in PTHS model mice can be improved by HDAC inhibitors through normalization of synaptic plasticity.

### 7.3. Rett Syndrome

Rett syndrome (RTT) is an unusual CVI in which visual impairment is mainly caused by defects in the brain instead of dysfunctions in the eyes [312]. Patients have difficulty recognizing unfamiliar objects [226] and their vision is selectively focused on specific objects [227]. It is difficult for them to search for specific objects in the visual field because they are unable to distribute their attention across it or shift attention from the distractors [228].

RTT is a rare neurodevelopmental disorder with normal initial development caused by loss-of-function mutations in the X-linked gene, methyl CpG binding protein 2 (MECP2) (Table 2), which encodes a DNA and histone methylation reader [143] with both transcription repressive and activating functions mediated through interactions with different cofactors [313]. In addition to methylated DNA and histones, MeCP2 also binds to RNA and is involved in mRNA splicing, miRNA processing, and other non-coding RNA-associated processes. Despite being an intrinsically disordered protein with a low content of secondary and tertiary structures [314], MeCP2 can be divided into several functional domains, namely N-terminal (NTD), methyl binding (MBD), intervening (ID), transcription repression (TRD), NCoR interaction (NID), and C-terminal (CTD) [315]. Such structural organization facilitates MBD-dependent binding to methylated DNA and is essential for interaction with transcriptional repressor mSin3A [316], nuclear receptor corepressor (NCoR) [317], Ski [318], putative Xenopus protease p20 [319], and DNA methyltransferase DNMT1 [320]. TRD is a secondary structure regulator recruitment platform for Ski [318], Ets family transcription factor PU.1 [321], splicing factors formin-binding protein (FBP) [322], Brahma (Brm) [323], RNA [324], and Y box-binding protein 1 (YB-1) of RNA splicing machinery [325]. NID functions to recruit the NCoR1/2 co-repressor complex to methylated DNA [326] but the targeted genomic sites remain unknown [327]. Most studies focus on mutations in NTD and MBD because of their importance for DNA and RNA binding [328]. NTD modulates the interaction with DNA [329] and influences the turnover rate of MeCP2 [330]. Although NTD of longer MeCP2-E1 isoform is encoded from exon 1 and NTD of shorter MeCP2-E2 isoform is encoded from exon 2, mutations in exon 1 would also reduce translation of MeCP2-E2 [331]. Insertions and deletions in exon 1 mainly occur at the polyalanine and polyglycine regions which are encoded from polyGGC and polyGGA stretches, respectively. Missense mutations are rarely reported but cause clinical severity. A2V mutation affects co- and post-translational modifications, reducing N-acetylation and polyalanine, eventually causing higher proteasomal degradation [330]. A59P mutation affects the overall conformation of the protein backbone, which influences MBD expression [332]. There has been only one RTT patient with mutations in exon 2 ever reported [333], so more studies are needed to confirm the pathological mechanisms.

Since MBD is the only domain with a well-defined tertiary structure in MeCP2, it is solely responsible for methylation-specific binding [315], which affects the stability [334] and affinity [333] of DNA and RNA binding. Although the change in folding free energy caused by missense mutations is small in absolute magnitude, ranging from a fraction of a kcal/mol up to more than 1 kcal/mol, the effect is significant because the total folding free energy in wild type is only about 2 kcal/mol [335]. Making up 45% RTT cases [336], missense mutations in MBD can be divided into three categories based on the stability of protein structure, namely reduced binding affinity with less stable structures (such as L100V, S134C, P152R, and D156E), reduced binding affinity without structural change (such as R106W, R106Q, R133H, R133C, F155S, T158M, and T158A) [337], and reduced binding affinity with more stable but less flexible structures (such as R111G and A140V) [338]. 

MeCP2 is expressed in all tissues but reaches near-histone abundance in neurons (~16 × 10^6^ molecules per nucleus) [338]. While MeCP2 mostly binds to chromatin and localizes within highly nuclease-accessible regions, a fraction of MeCP2 molecules can loosely bind to the nucleosome-depleted regions [143]. Moreover, several studies have shown that MeCP2 can regulate alternative splicing, which is a strict requirement for almost all neurotransmitter receptors and channels [325]. Although the molecular mechanisms of genetic disruptions in brain and eye functions remain obscure [334], it is believed that MeCP2 affects the maturation of the CNS because it is expressed earlier in the ontogenetically older structures, such as the brainstem, than in the newer structures, such as the hippocampus or cerebral cortex [339]. Neurons seem to be less mature in RTT patients based on the decrease in brain size [340], neuronal size [341], and dendritic branching instead of neuron number [342]. Therefore, patients with RTT display microcephaly [343], seizures [344], and developmental regression in speech and hand skills after initially normal development [345].

For disease management, although there have not been any FDA-approved treatments for RTT, gene therapy experiments in mice showed restoration of MeCP2 reverses pathology even at adult stages (Table 5) [346]. Vectors for gene therapy are mainly based on retroviruses, including lentivirus and adeno-associated virus (AAV) [347]. However, lentivirus cannot cross the blood–brain barrier (BBB) and its spread beyond the injection site is limited [348], so AAV is a better choice for nervous system disorders [349] as it can mediate long-term transgene expression [350]. Nonetheless, it should be noted that AAV-mediated gene delivery may cause toxicity due to the uncontrolled expression level of the transgene [351], and any overexpression of MeCP2 would impair brain functions [352]. Given that RTT is a disorder caused by the lack of neuronal maturation, treatment during infancy could improve the effects and reduce the amount of treatments required. In addition, since the deletion of MeCP2 in mature neurons is deleterious [353], long-lasting treatment effects are required. CRISPR/Cas9-mediated mutation correction may be a potential treatment [354], but more studies are still needed in order to reach clinical requirements.

### 7.4. Cockayne Syndrome

Cockayne syndrome (CS) is an autosomal recessive multisystem degenerative disorder caused by the mutations in two complementary genes, namely the excision repair cross-complementation group 6 (*ERCC6*) (80%) [144] and the excision repair cross-complementation group 8 (*ERCC8*) (20%) (Table 2) [145]. ERCC6, also known as Cockayne syndrome B (CSB), is a member of the SWI2/SNF2 family of chromatin remodeling complexes [368], involved in mitochondrial DNA (mtDNA) damage repair, base excision repair (BER), interstrand crosslink (ICL) repair, and double-strand break (DSB) repair. All of the components in the mitochondrial transcription apparatus, including mitochondrial RNA polymerase, transcription factor 2B, and mitochondrial transcription factor A (TFAM), can stimulate ATPase activity of CSB, which is required for mtDNA repair [369]. CSB stimulates the BER pathway to repair oxidatively-induced DNA damage by interacting with poly(ADP-ribose) polymerase 1 (PARP-1) [370] or apurinic/apyrimidinic (AP) endonuclease (APE1) [371]. CSB stimulates the exonuclease activity on single- and double-stranded oligonucleotides of nitrogen mustard 1A (SNM1A) to unhook ICL [372]. The ATPase activity of CSB abrogated by mutations impairs [373] homologous recombination (HR) caused by BRCA1 and nonhomologous end joining (NHEJ) promoted by 53BP1 and Rif1 [374]. Although mutations in CSB impair DNA repair, cells with CSB mutations prematurely enter the G2/M stage of the cell cycle because of the reduction in DNA damage responses mediated by ataxia telangiectasia mutated (ATM) and checkpoint kinase 2 (CHK2) [374].

The Cockayne syndrome A (CSA) protein encoded by *ERCC8* belongs to the WD40 repeat family, and binds to CUL4 of the cullin ring ubiquitin ligase complex (CRL4) through the DDB1 adaptor to regulate DNA repair via ATF3, CSB, and p53 [375]. The removal of ATF3, the product of an immediate early gene (IEG), prevents the recruitment of RNA polymerase (Pol II) to DNA damage sites, which blocks the restart of RNA synthesis [376]. Moreover, CSB is another CRL4CSA ubiquitination target, which reversely releases the inhibition of CRL4CSA by the COP9 signalosome complex (CSN) [377]. CRL4CSA as well as CRL4CSB can also stimulate the ubiquitination of tumor suppressor p53 [378], which mediates the balance between the removal of highly damaged cells via apoptosis [379] and the survival of slightly damaged cells after proper repair [380]. By establishing a negative feedback loop, p53 can transcriptionally affect CSB via binding to the promoter region, which results in the maintenance of a steady level of p53. 

Clinically, CS spans a phenotypic spectrum of severity, including Cockayne syndrome type I (CS type I), Cockayne syndrome type II (CS type II), and Cockayne syndrome type III (CS type III) [230]. CS type I is a moderate and the most prevalent (85% of cases) [60] form, with normal prenatal growth and 16.1 years of mean age at death [55], mainly caused by mutations in CSA [60]. CS type II is a severe and early-onset form with growth failure at birth and five years of mean age at death, mainly caused by CSB, which participates in more DNA repair pathways [60]. Cerebro-oculo-facio-skeletal (COFS) syndrome is a more severe subtype of CS type II with the presence of arthrogryposis [381]. CS type II is a mild and late-onset form with symptoms occurring from two years after birth, with a mean death age of 30.3 years [60]. Premature ageing caused by defects in DNA repair lead to ageing-related abnormalities, such as neurodevelopmental defect and loss of retinal cells [382]. Therefore, patients with CS selectively exhibit abnormal myelination in the brain (93%), intracranial calcifications (63%) [61], epilepsy (5–10%) [230], pigmentary retinopathy (60–100%), and cataracts (15–36%) [383].

Additionally, some CS patients exhibit the combined phenotype with xeroderma pigmentosum neurological disease (XP/CS) [61]. While XP is primarily a neurodegenerative disease caused by defects in the nucleotide excision repair (NER) system, and CS appears to be a neurodevelopmental disease, clinical features of patients with XP/CS intertwine. There are seven XP complementation groups: XPA, XPB, XPC, XPD, XPE, XPF, XPG, and XPV. XP/CS is often associated with XPB (encoded by *ERCC3*), XPD (encoded by *ERCC2*), XPF (encoded by *ERCC4*), and XPG (encoded by *ERCC5*) [61]. Patients with XP/CS are susceptible to acute sunburns after minimum exposure and are likely to have facial freckling or pigmentary changes that are uncommon in pure CS [61].

In terms of disease management, there are currently no FDA-approved drugs or curable treatments for CS. Since survival beyond childhood is unusual, the main goal of management is to maximize quality of life. Presently, there is one study attempting to correct genes with CRISPR/Cas9 in iPSCs reprogrammed from the fibroblasts of a CS patient, and the results showed some recovery of DNA repairability (Table 5) [360]. More studies are needed for better management of CS. 

### 7.5. X-linked Alpha Thalassaemia Mental Retardation

X-linked alpha thalassaemia mental retardation (ATR-X) syndrome is a rare human congenital X-linked recessive neurodevelopmental disease primarily affecting males. ATR-X presents with a wide range of symptoms, including developmental impairment, intellectual disability, growth impairment, gastrointestinal manifestations, genital anomalies, hypotonia, seizures [384], and ocular defects [241]. Susceptible loci related to the neurodevelopmental disease are mostly involved in chromatin or transcriptional regulation, including chromatin remodelers altering nucleosome spacing or facilitating histone variant exchange using energy from ATP hydrolysis [385]. The *ATRX* locus encodes two major transcripts encoding transcriptional regulators, one full-length protein and one truncated isoform lacking an ATP-dependent remodeling domain [146]. ATR-X and death domain associated protein (DAXX) interact with each other via the regions of ATRX-DNMT3-DNMT3L (ADD) and SNF2-ATPase domains. ATRX-DAXX deposits the histone variant H3.3 at pericentric and telomeric repeats [386] mediated by promyelocytic leukemia protein (PML) [387] to the heterochromatin histone mark, H3K9me3, by either indirectly interacting with heterochromatin protein 1 (HP1α) [388] or methyl-CpG-binding protein (MeCP2) [389], or by directly binding to H3K9me3 with ADD [390]. H3.3K9me3 recruits more ATRX-DAXX-H3.3 complexes, which creates a positive feedback loop to silence non-coding telomeres [391]. Recently, several studies showed that ATRX promotes telomere cohesion between sister telomeres to mediate the repair of double-strand DNA breaks [392]. In brief, ATRX regulates the transcription of telomeres via histone variants.

Patients with ATR-X syndrome have reduced ATRX protein levels [393] which are mainly caused by missense mutations in the ADD domain (50%) [394] and SNF2-ATPase (30%) [395]. ADD mutations occur in the N-terminus of both transcripts and reduce localization to chromocenters [393], whereas SNF2-ATPase mutations only occur in the C-terminus of full-length protein and cause attenuation of ATPase activity and the reduction of localization to PML nuclear bodies [396]. 

In ATRX-null cells, the increase of telomeric repeat-containing RNA (*TERRA*) enhances the formation of RNA-DNA hybrids (R-loops) and stabilizes G-quadruplex secondary DNA (G4 DNA) [397], which are the structures formed by tandem repeat DNA elements such as variable number tandem repeats (VNTRs) [398]. Both R-loops and G4 DNA structures can recruit ATRX to re-establish a normal chromatin structure, but they cannot be resolved effectively in the absence of ATRX [397]. Delayed cell-cycle progression is observed in the S and G2/M phases of the cell cycle [399]. In the mid-late S-phase, genomic instability is enriched at telomeres and pericentromeric heterochromatin [399] and in the G2/M phase, sister chromatid cohesion and congression defects affect separation at anaphase [400]. To sum up, the loss of ATRX causes delay cell cycle and genomic instability.

In ATR-X syndrome, microcephaly occurs in 75% of the cases [233] and ocular defects are present in ~25% of the cases (Table 3) [234]. The delayed cell cycle reduces the stability of the neural progenitor cell (NPC) pool, which leads to a reduction in upper layer neurons and a decrease in brain size [401]. Therefore, most ATR-X patients develop postnatal microcephaly [233] and patients with a higher number of variant number tandem repeats exhibit more severe α-thalassemia [402]. An in vivo study also found that a reduced ATRX protein level leads to the loss of amacrine and horizontal cells [234] and defects in retinal bipolar cells by affecting post-replicative neuronal integrity in the CNS [403]. Transcriptional deficits associated with ATRX mutations are the main molecular pathological mechanism of ATR-X syndrome.

For clinical management, so far there are no FDA-approved therapies for ATR-X syndrome. 5-aminolevulinic acid (5-ALA) can be a potential therapeutic strategy to target G4 DNA, which has shown promising results in ATR-X model mice [404] and Japanese patients [405]. However, more studies are required.

## 8. Compromised Peroxisomes

Cellular stress arises from either the overloading of misfolded proteins or the clearance of malfunctioned peroxisome debris (Figure 1). Peroxisome component gene mutations have been found to cause diseases such as Zellweger spectrum disorders (ZSDs) including Refsum disease. 

### 8.1. Zellweger Spectrum Disorder

Zellweger spectrum disorders (ZSDs) are a group of diseases that include such disease entities as Zellweger syndrome (ZS), neonatal adrenoleukodystrophy (NALD), and infantile Refsum disease (IRD). The diseases result from different gene mutations but have overlapping clinical presentations, with ZS being the most severe and IRD the least severe form. Individuals with ZS typically do not survive past the first year after birth and develop hypotonia, cataracts, nystagmus, seizures, renal and hepatic problems, and craniofacial dysmorphia in their lifespan. NALD patients can survive until teenage years and IRD patients even until adulthood, however, symptoms such as developmental delay, hypotonia, chorioretinopathy, sensorineural hearing loss, and hepatomegaly still emerge in late infancy and persist in the long term [406,407]. 

The *PEX* gene family encodes peroxins whose functions are crucial for peroxisome biogenesis and peroxisomal transport, and the mutations in *PEX* genes are often implicated in ZS [147]. The most prevalent *PEX* mutations occur in *PEX1* (58.9%), *PEX6* (15.9%), and *PEX12* (7.1%) genes. *PEX1* and *PEX6* encode cytosolic AAA ATPase protein family members, while *PEX12* encodes peroxisomal membrane protein [408]. PEX1 and PEX6 form a heterodimer with ATPase activity by interacting via their C-terminal nucleotide-binding domains. The ATPase activity is required for translocation and unfolding the substrates for further hydrolysis. It has been demonstrated that yeast growing on oleic acid media requires the intact pore loop of the PEX1/6 D2 domain to execute oleic acid β-oxidation within the peroxisome [409]. In mammals, the branched and very long chain fatty acids (VLCFAs, C > 22) are catabolized mostly in mitochondria, but the proper peroxisome function is required to clear out the excessive VLCFA metabolites. In ZSDs, the *PEX* mutations often jeopardize the peroxisomal function in β-oxidation of the VLCFAs. Docosahexaenoic acid (DHA) is one of such fatty acids, and the biogenesis and catabolizing of DHA are crucial for neuronal health, as its deficiency is associated with visual impairment [410]. 

The accumulation of VLCFA and the absence of plasminogen synergistically modulates gliosis, inflammation, and axonopathy in *Pex7:Abcd1* double KO mice, resulting in tremors and hindlimb ataxia [411]. To summarize, altering such pathways as β-oxidation of methyl-branched fatty acids (e.g., pristanic acid), dihydrocaffeic acid (DHCA), tetrahydrocannabinolic acid (THCA), α-oxidation of fatty acids such as phytanic acid [412], fatty acid racemization, ether phospholipid (plasmalogen) biosynthesis, detoxification of glyoxylate, and reactive oxygen species is associated with both brain damage and visual impairment [413]. 

### 8.2. Refsum Disease

Refsum disease (RD) is a rare autosomal recessive hereditary motor and sensory neuropathy type IV caused by deficient oxidation of phytanic acids (PA) [414]. Despite the polyisoprenoid-like structure of PA, in vivo studies in humans and animals indicate that PA is derived only from dietary sources, especially from dairy products and ruminant fats, instead of endogenous synthesis, in a way similar to the synthesis of isoprenoids from acetate via the mevalonate pathway [415]. Although the mechanisms of the elevated levels of PA resulting in disease remain unknown, several susceptibility genes have been discovered [416]. 

There are two subtypes of RD: adult Refsum disease (ARD) [148] and infantile Refsum disease (IRD) [417]. ARD is caused by the deficiency of phytanoyl-CoA hydroxylase (PAHX) encoded by *PHYH* (90%) or the type 2 peroxisomal targeting signal (PTS2) receptor encoded by *PEX7* (10%) (Table 2). PAHX requires 2-oxoglutarate, Fe^2+^ and ascorbate to catalyze the first step in the α-oxidation of PA [418] in peroxisomes [419]. PAHX is a typical PTS2 protein with an Xn–RL–X5–HL–Xn consensus motif near the N-terminus to be recognized by the PTS2-receptor. PAHX-PTS2 receptors are then recognized by the proteins on the peroxisomal membrane, followed by the translocation of PAHX across the peroxisomal membrane and the recycling of the PTS2 receptors back to the cytosol [415]. IRD is associated with the mutations in at least 12 different *PEX* class genes [420], including *PEX1*, *PEX2*, and *PEX26* encoding ATPases, which import cytosolic proteins into peroxisomes [69]. In addition to PA, accumulation of other substrates, primarily VLCFA and di- and tri-hydroxycholestanoic acid, and pipecolic acid, also occur in IRD [420]. Therefore, IRD is more severe, with the onset occurring in early infancy [67] and a survival of only 5–13 years [69], while ARD is milder with the onset in 2–7 years [67] and a survival of 4–5 decades [69].

The main clinical features of RD include ophthalmology (100%) and polyneuropathy (70%) (Table 3) [192]. Retinitis pigmentosa (RP) with constricted visual fields and night blindness was observed in all patients [421], mostly before additional clinical symptoms [68]. Similar to other forms of rod-cone dystrophy, cataracts often develop earlier than age-related cataracts, but surgery may be constrained by poor pupillary dilatation caused by atrophy of the iris dilator muscle [417]. Polyneuropathy is a chronic and progressive mixed-motor and sensory type, eventually leading to muscular atrophy and weakness [69]. Additionally, protein levels increase in the CSF without an increase in the number of cells [422]. The main purpose of treatment is to reduce PA levels in plasma and tissue. Dietary restriction in PA, including beef, lamb, and dairy products, is recommended to be 10–20 mg/day compared to the normal average intake of 50–100 mg/day [414]. In order to achieve a rapid and significant decrease in PA levels, plasma exchange should also be taken into consideration [423]. Although PA levels are not normalized completely in most patients, the reduction of PA still shows definite clinical improvement [424].

## 9. Channelopathies

Channelopathies, including neuromyelitis optica spectrum disorder (NMOSD), are a group of disorders of the nervous system resulting from the dysfunction of ion channels. NMOSD is an autoimmune disease characterized by optic neuritis (ON), longitudinally extensive transverse myelitis (LETM), area postrema syndrome, and acute brainstem syndrome, affecting the optic nerve and spinal cord (Table 3). Compared to MS, ON is more likely to have initial simultaneous bilateral manifestations, recurrence, and poorer long-term visual outcomes [425]. LETM is often accompanied by profound bilateral motor weakness, prominent dysesthesias, and sensory level sphincter dysfunction [426]. Area postrema syndrome features nausea [427], and brainstem syndromes often include vomiting, hiccups, facial nerve palsy, oculomotor dysfunction, and vertigo [237]. Because NMOSD was once regarded as a subtype of MS, early attempts to find its susceptibility loci focused on the human leukocyte antigen (HLA) region. However, various studies have found different susceptibility alleles between MS and NMOSD in *HLA-A* [167], *B* [167], *C* [168], *DPB1* [80], *DRB1* [168,169,170,171,428], *DQA1* [168,172], *DQB* [171], and *DQB1* (Appendix A) [171,174]. Eventually, NMOSD was recognized as a distinct disease because of the discovery of antibody biomarkers (NMO-IgG) [429]. Identified within NMO-IgG, pathogenic water channel aquaporin 4 (AQP4) antibodies, a T cell-dependent immunoglobulin subclass (IgG1), were detected in 60–90% of NMO patients [430]. AQP4 is expressed mostly in the astrocytes of the CNS which happen to be at the interface of blood vessels and neuron systems [431]. Moreover, AQP4 is also expressed in the supportive Müller cells of the retina [432]. After entering the CNS from plasma [433] or secreted by plasma cells in the CSF [434], AQP4-IgGs bind preferentially to orthogonal arrays of particles (OAPs), the supramolecular assemblies of AQP4 tetramers, and initially lead to complement-dependent cytotoxicity (CDC) under the presence of complement or antibody-dependent cellular cytotoxicity (ADCC) under the presence of the effector cells, such as natural killer cells (NK cells) [435]. Subsequent astrocyte cytotoxicity caused by inflammatory events, such as granulocyte infiltration or macrophage infiltration [436], leads to the loss of AQP4 and the astrocyte marker glial fibrillary acidic protein (GFAP) as well as the disruption of the blood–brain barrier (BBB), followed by oligodendrocyte and neuronal cell death. Therefore, SNPs in *AQP4* [156] or T cell marker genes (such as *CD58* [159,162,163,164], *CD127* [165], *CD226* [166], and *NECL2* [155]) and susceptible loci related to complement system (such as *CFB* [149] and *C4B* [175]) or NK cell markers (such as *PRF1* [151]) can be associated with susceptibility to NMOSD. 

Approximately 90% of NMOSD patients exhibit a relapsing course: 50% within 1 year and 90% within 5 years, with the risk not diminishing with age [437]. Therefore, long-term immunotherapy is usually conducted soon after diagnosis to avoid disability accrual. There are only three treatments for NMOSD approved by the FDA, but only for adult patients with AQP4-IgG+, including a terminal complement protein (C5) inhibitor Soliris (eculizumab) injection in 2019 [438], an afucosylated IgG1 kappa monoclonal antibody Uplizna (inebilizumab-cdon) injection targeting CD19 [439], and a humanized monoclonal antibody Enspryng (satralizumab-mwge) targeting the interleukin-6 (IL-6) receptor in 2020 [436]. Additionally, monoclonal antibody rituximab-targeting CD20 [440], purine analog azathioprine-blocking deoxyribonucleic acid synthesis during B and T cell proliferation [441], and mycophenolate mofetil (MMF), the prodrug of mycophenolic acid [442], have been widely used off-label to prevent relapses. Tocilizumab, the first humanized anti-IL-6 receptor monoclonal antibody, has demonstrated safety and efficacy in an open-label, multicentre, randomized phase 2 trial [443]. Currently, an open-label phase I clinical trial is ongoing, using T lymphocytes with genetic modification of chimeric antigen receptors (CAR) targeting B cell maturation antigen (BCMA) in AQP4-IgG-positive patients, and the first results are expected in 2023 ([361]). However, since the molecular mechanisms of pathology in AQP4-IgG-negative disease remain unclear [444], initiation of immunosuppressive therapy is recommended [445].

## 10. Conclusions

CVIs are a type of comorbidities affecting both visual and CNS functions due to common mechanisms underlying the functionality of the eye and the brain. While CVIs are often underdiagnosed, the genetic background underlying CVIs deserves more clinical attention and advanced technology investment. Fortunately, the transactions between GWAS and PheWAS fostered big data that enables physicians and scientists to revisit the genetics aspect and to develop intervention strategies for CVIs. With this review, readers could acquire the multifaceted perspectives of the pathomechanisms in these rare diseases.

## Figures and Tables

**Figure 1 ijms-23-09707-f001:**
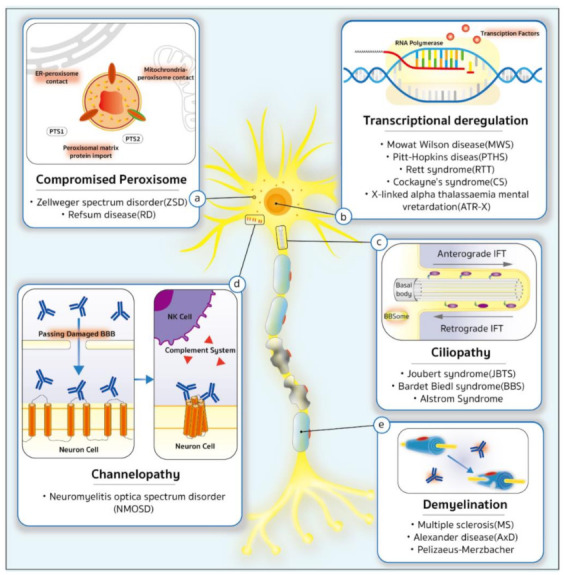
Five categories of pathology of cerebral visual impairments. (**a**) Compromised peroxisome diseases occurring because of deformation of peroxisomes or failure in peroxisomal protein transportation; (**b**) transcriptional deregulation diseases resulting from mutant transcription factors; (**c**) ciliopathies caused by instability of transportation and structure in the cilia; (**d**) Channelopathies resulting in cytotoxicity; (**e**) demyelinations associated with abnormal immune systems.

**Figure 2 ijms-23-09707-f002:**
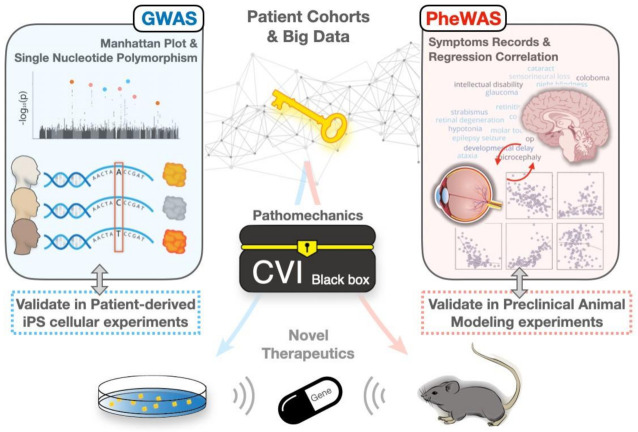
Revisiting cerebral visual impairments (CVIs) by genome-wide and phenome-wide association studies (GWAS and PheWAS). Cerebral visual impairments result from hidden common pathomechanisms. With the transactions between GWAS and PheWAS, hidden pathologies could be revealed and further proved by studies in induced pluripotent stem (iPS) cell and animal models, which could stimulate the inventions of novel therapeutics.

**Table 2 ijms-23-09707-t002:** Pathomechanisms of Cerebral Visual Impairments.

Type	Disorder ^1^	Phenotype OMIM ^2^ Number	Subtypes	Gene or Susceptibility Locus	Chromosomal Location	Gene OMIM ^2^ Number	Protein	Molecular Level	Reference
**Ciliopathy**	JBTS^2^	See in Appendix A						Transition zone (TZ) SHH signaling basal body (BB)	[88,89,90,91,92,93,94,95,96,97,98,99,100,101,102,103,104,105,106,107,108,109,110,111,112,113,114,115,116,117,118,119,120,121,122,123,124,125,126,127,128,129,130,131]
	BBS^2^	See in Appendix A						BBSome protein chaperonin complex IFT	[132,133]
	Alstrom syndrome	203,800	-	*ALMS1*	2p13.1	606,844	ALMS1	unclear	[134]
**Demyelination**	MS ^2^	126,200	MS1	*HLA-DQB1* *HLA-DRB1*	6p21.32	604,305 142,857	HLA class II histocompatibility antigen, DQ beta 1 chain HLA class II histocompatibility antigen, DRB1 beta chain	chronic inflammation	[79,135,136,137]
		612,594	MS2	*MS2*	10p15.1	612,594	-		
		612,595	MS3	*MS3*	5p13.2	612,595	-		
		612,596	MS4	*MS4*	1p36	612,596	-		
		614,810	MS5	*TNFRSF1A*	12p13.31	191,190	Tumor necrosis factor receptor superfamily member 1A		
	AxD ^2^	203,450	-	*GFAP*	17q21.31	137,780	Glial fibrillary acidic protein	GFAP aggregates	[138,139]
	PMD ^2^	312,080	-	*PLP1*	Xq22.2	300,401	Myelin proteolipid protein	PLP1 accumulation	[140]
**Transcriptional Deregulation**	MWS ^2^	235,730	-	*ZEB2*	2q22.3	605,802	Zinc finger E-box-binding homeobox 2	transcription repressor targeting 5′-CACCT sequences interaction with Smads, TGFβ, and NuRD complex	[141]
	PTHS ^2^	610,954	-	*TCF4*	18q21.2	602,272	Transcription factor 4	transcription of neurogenesis	[142]
	RTT ^2^	312,750	-	*MeCP2*	Xq28	300,005	Methyl-CpG-binding protein 2	DNA and histone methylation reader transcription factor	[143]
	CS ^2^	-	CS type I	*ERCC6* *ERCC8*	10q11.23 5q12.1	-	CSB CSA	DNA repair	[144,145]
			CS type II						
			CS type III						
			XP/CS	*ERCC2* *ERCC4* *ERCC5*	19q13.32 16p13.12 13q33.1	-	General transcription and DNA repair factor IIH helicase subunit XPD DNA repair endonuclease XPF DNA excision repair protein XPG		
	ATR-X ^2^	301,040	-	*ATRX*	Xq21.1	301,040	Transcriptional regulator ATRX	depositting histone variant	[146]
**Compromised Preoxisome**	ZSD ^2^	-	-	*PEX1~13*	-	-	-	peroxisome formation peroxisomal protein transport	[147]
	RD ^2^	-	ARD	*PHYH* *PEX7*	10p13 6q23.3	-	phytanoyl-CoA hydroxylase type 2 peroxisomal targeting signal receptor	oxidation of phytanic acid	[148]
			IRD	*PEX1~13*	-	-	-		
**Channelopathy**	NMOSD ^2^	-	-	See in Appendix A				cytotoxicity related to T cell, complement, NK	[79,149,150,151,152,153,154,155,156,157,158,159,160,161,162,163,164,165,166,167,168,169,170,171,172,173,174,175]

^1^ OMIM—Online Mendelian Inheritance in Man. ^2^ JBTS—Joubert syndrome; BBS—Bardet–Biedl syndrome; MS—multiple sclerosis; AxD—Alexander disease; PMD—Pelizaeus–Merzbacher disease; MWS—Mowat–Wilson disease; PTHS—Pitt–Hopkins disease; RTT—Rett syndrome; CS—Cockayne syndrome; ATR-X—X-linked alpha-thalassaemia mental retardation; ZSD—Zellweger spectrum disorder; RD—Refsum disease; and NMOSD—neuromyelitis optica spectrum disorder.

**Table 5 ijms-23-09707-t005:** Research in Gene Therapy.

Disease ^1^	Target Gene	Mutation	Cas9 Ortholog and Delivery	Editing Mechanism	Model	Main Results	Reference
**BBS1**	*BBS1*	M390R	AAV2/5 vectors	Insert between two ITRs	^M390R/M390R^ mice	24% to 32% transduction in retina higher b-wave amplitudes in 50% mice	[263]
**RTT**	*MeCP2*	-	AAV9	-	*Mecp2* null mice	Transduction efficiency: ~2–4% neurons observed improvements in survival	[351]
	*MeCP3*	-	AAV9	-	*Mecp2* null male mice	Partial amelioration in the null mouse model via provision of exogenously derived MeCP2	[351]
	*MeCP4*	-	scAAV9	-	*Mecp2* stop mice *Mecp2B* null mice	Reversing symptoms by ectopic expression of MeCP2 in virus infecting peripheral tissue and multiple cell types within the CNS	[355]
	*MeCP5*	-	AAV9	-	*Mecp2* KO mice	Mecp2 transgene correcting breathing deficits and improving survival	[356]
	*MeCP6*	-	AAV9	-	*Mecp2* null mice *Mecp2tm1.1Bird* mice *Mecp2T158M* mice	Direct cerebroventricular injection into neonatal mice resulting in high transduction efficiency, increased survival and body weight, and an amelioration of RTT-like phenotypes	[357]
	*MeCP7*	-	AAV9	-	*Mecp2*^−/y^ mice	Modified vector extending lifespan without rescuing behavior	[358]
	*MeCP8*	R270X	SpCas9, T2A	repairing induced DSBs by HR	*MECP2R270X* iPSC	developing CRISPR/Cas9-mediated system modifying *MECP2* locus	[354]
	*MeCP9*	-	AAV9	-	*Mecp2*^−/y^ mice	Insertion of miRARE improving safety without compromising efficacy	[359]
**CS**	*ERCC6*	c.643G>T (p.E215X)	pCAG-mCherry-gRNA vector	replace mutation with ssODN	CS-iPSCs	Alleviation of aging defects and recovered DNA repair ability	[360]
**NMOSD**	*CART-BCMA*	—	lentiviral vector	Add scFv	-	-	[361]
**adRP**	*Rho*	S334ter	SpCas9, plasmid electroporation	Allele-specific knockdown by indel	*S334ter-3* rats	Nine-fold increase in photoreceptor nuclei 53% Improvement in the optokinetic response	[362]
	*Mertk*	1.9 kB deletion (intron 1–exon 2)	SpCas9, two AAV8 or 9 vectors	HITI-mediated insertion	*RCS* rats	Electroretinogram showing improved rod and cone responses compared with untreated and HDR-treated controls	[363]
	*Nrl*	—	SpCas9, two AAV8 vectors	Knockdown by indel (reprogram rods to cone-like cells)	*Rho^−/−^* and *rd10* mice	25% increase in cone photoreceptor preservation and electroretinogram B waves amplitude by ~60%	[364]
	*Rho*	P23H	SpCas9, two AAV8 vectors	Allele-specific knockdown by indel and wild-type supplementation	*Rho^P23H/P23H^* and *Rho^P23H/+^* knock-in mice	Preserved electroretinogram B-waves and outer nuclear layer thickness in Cas9-treated mice compared with mice only given gene supplementation	[365]
	*Rho*	P23H	SpCas9-VQR, plasmid electroporation	Allele-specific knockdown by indel	*Rho^P23H/+^* knock-in mice	Increase in wild-type mRNA by ~20% compared with untreated control Delayed outer nuclear layer degeneration	[366]
	*Pde6b*	Y347X	SpCas9/RecA, plasmid electroporation	Induce HDR using sgRNA-targeted RecA	*rd1* mice	Increased survival of rod photoreceptors five-fold compared with nontreated controls	[367]

^1^ BBS—Bardet–Biedl syndrome; RTT—Rett syndrome; CS—Cockayne syndrome; NMOSD—neuromyelitis optica spectrum disorder; and adRP—autosomal dominant retinitis pigmentosa.

## Data Availability

Not applicable.

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
