# Peer review of "Genetics behind Cerebral Disease with Ocular Comorbidity: Finding Parallels between the Brain and Eye Molecular Pathology"

_ijms, 2022, doi:10.3390/ijms23179707_

Round 1
Reviewer 1 Report
The manuscript of Kao-Jung et al. is aimed at providing a comprehensive review concerning parallels between brain and ocular molecular pathologies. Authors complement the manuscript with 3 well designed Figures, 4 Tables and 3 Supplementary Tables that summarize previous studies/publications. Authors cite more than 500 publications and provide an exhaustive review of this complex topic from different angles.
The manuscript fits the scope of the “Journal of Molecular Sciences” and is of interest for the readers of the journal. The language of the manuscript needs minor revision, in particular for a more scientific language (see some examples in the points detailed below).
The Figures are well designed, however, Authors should revise Figure and Table Legends and provide the abbreviations at the end of each legends to make them understandable by themselves.
There are some stylistic and presentation issues that should be addressed before it could be considered for acceptation.
Major points of concern:
1. In general, English usage should be revised for more scientific style: e.g.
- Page 1-2: ”These young patients typically present with visual difficulties that cannot be explained by ophthalmological examinations, and in some somatic mutation cases, their condition were not traceable in their pedigree, which generally makes the characterization of CVI challenging and clueless.” Clueless is a strange word to use in this context.
- Page 2 “To this point, there exists no standard approach of how and what to investigate in the CVIs”. There ARE existing guidelines concerning the investigations and diagnosis, however, currently no consensus exists. Please rephrase the sentence.
- Page 5 “and so on”. Authors should list all individual gene mutation types, or use the phrase “and several other types” and include a review reference about this specific topic.
This is not an exhaustive list of non-scientific phrases, please have the text revised for such stylistic errors.
2. Table 1: Please revise, the text in the columns, they do not correctly fit the space. It makes the Table difficult to read. Please list ALL abbreviations in the Table Legend (even the ones already defined in the text, the Table should be readable in its own).
3. Table 2: Please see remarks about Table 1.
4. Table 3: please see remarks about previous Tables. Table 3 has specifically problems with Column 1 (Type).
5. Figure 3 should rather be named Table 4.
6. Table 4: see remarks concerning previous Tables. There are some parts with different font type than the majority of the text, please correct. No need for the column “Year and Month”, please delete.
Reviewer 2 Report
Review of a manuscript “Genetics behind Cerebral Disease with Ocular Comorbidity: Finding Parallels between the Brain and Eye Molecular Pathology” by Kao-Jung Chang and coauthors.
It is well known that some of the cerebral visual impairments include a wide spectrum of diseases with various visual defects with associated genetic brain disorder. The authors summarize the data dealing with genetic factors background and molecular functions at the cellular level that result in the consequential brain and eye defects in cerebral visual impairments. This is an important field of biomedical research and the results presented in the manuscript will be interesting for the readers of the journal.
The following corrections and addition should be made.
-The Genetic Predisposition to CVIs. “Most genetically-driven CVIs are rare diseases with incidence ranging from 1/90,000 to 1/2,700,000 (Table 1), so these genetically caused CVIs are less prioritized for scientists or pharmaceutical companies to invest for solutions.” This is a clumsy sentence which should be corrected as follows:” Most genetic CVIs are rare diseases with incidence ranging from 1/90,000 to 1/2,700,000 (Table 1). Thus, these types of disorders are less attractive for the investment by pharmaceutical companies in order to study their mechanisms and develop treatment”
-After the sentence “Diagnoses, pathologies, and treatments of these diseases remain largely unknown, which compromises the right of patients to live a healthier life” the authors should add the following text and citations: “The association between visual dysfunction and brain diseases was found in many previous studies [references: 1 Maurage CA, et al. Retinal involvement in dementia with Lewy bodies: a clue to hallucinations? Ann Neurol. 2003; 54 (4):542-7. doi: 10.1002/ana.10730. PMID: 14520672.
2 Conformational diseases: looking into the eyes. Brain Res Bull. 2010 Jan 15;81(1):12-24. doi: 10.1016/j.brainresbull.2009.09.015. PMID: 19808079.]
-“Such mutations can affect individual genes, among them are missense mutations, reading frameshifts, nonsense mutations, exon deletions, truncations, alternative splicing mutations, terminal readthrough, and so on.” This bad style should be corrected as follows: ”These mutations can be classified as missense, frameshifts, or nonsense; other types of DNA alterations may appear as a result of exon deletions, truncations, alternative splicing, etc.”
“In such a way, the pathological events occurring in these rare genetic diseases can be connected with with common molecular pathways (Table 2)”
This should be corrected as follows:”Using this approach, the pathological changes occurring in these rare genetic diseases can be associated with common molecular pathways (Table 2)”
Bad style, hard to understand the sentence “With thorough understanding of these clinical features, doctors may diagnose and supply pre-treatment before the later symptoms appearance” should be corrected as follows:” Revealing molecular mechanisms underlying these clinical features, physicians will be able to diagnose the pathology and apply treatment before the appearance of symptoms”.
Conclusion
“Cerebral visual impairments (CVIs) “ this abbreviation was used in the very beginning, so it should not be used in Conclusion.
Overall, the manuscript is overloaded with details. It would benefit if the authors make it more concise.
Round 2
Reviewer 2 Report
The manuscript is improved in response to Reviewer's comment